# Germline Non-CDKN2A Variants in Melanoma and Associated Hereditary Cancer Syndromes

**DOI:** 10.3390/diseases13060180

**Published:** 2025-06-09

**Authors:** Chiara Anna Fiasconaro, Alice Carbone, Silvia Giordano, Francesco Cavallo, Paolo Fava, Barbara Pasini, Yuliya Yakymiv, Sara Marchisio, Pietro Quaglino, Simone Ribero, Gabriele Roccuzzo

**Affiliations:** 1Department of Medical Sciences, Section of Dermatology, University of Turin, 10126 Turin, Italy; chiaraanna.fiasconaro@unito.it (C.A.F.); alice.carbone98@gmail.com (A.C.); si.giordano@unito.it (S.G.); francesco.cavallo@unito.it (F.C.); fava_paolo@yahoo.it (P.F.); pietro.quaglino@unito.it (P.Q.); simone.ribero@unito.it (S.R.); 2Medical Genetics Unit, AOU ‘Città Della Salute e Della Scienza’, ‘Molinette’ Hospital, 10126 Turin, Italy; barbara.pasini@unito.it; 3Laboratory of Immunogenetics, Department of Medical Sciences, University of Turin, 10126 Turin, Italy; yuliya.yakymiv@unito.it (Y.Y.); sara.marchisio@unito.it (S.M.)

**Keywords:** melanoma, hereditary melanoma, germline variant, genetic test, genes, tumor associations, hereditary cancer syndromes

## Abstract

The etiology of melanoma is multifactorial and arises from the interplay of genetic, phenotypic, and environmental factors. The genetic predisposition to melanoma is influenced by a complex interaction among genes exhibiting varying levels of penetrance (high, moderate, and low), each contributing differently to the susceptibility of the disease. Furthermore, penetrance may vary based on the incidence of melanoma across diverse populations and geographical regions. Advances in genetic sequencing technologies have facilitated the identification of novel genes potentially associated with melanoma, as well as the characterization of relevant germline variants. While the most extensively researched variant is CDKN2A, recent studies have highlighted other variants unrelated to CDKN2A as significant areas of investigation. Among them, high-penetrance genes encompass CDK4, BAP1, POT1, TERT, ACD, and TERF2IP. In contrast, moderate-penetrance genes include MC1R, MITF, and SLC45A2, while low-penetrance genes consist of OCA2, TYRP1, and TYR. In addition to elevating the risk of melanoma, these genetic alterations may also predispose individuals to internal neoplasms. This review aims to provide a comprehensive overview of the definitions of sporadic, multiple primary, familial, and hereditary melanoma, with a particular emphasis on non-CDKN2A germline variants and their dermoscopic and phenotypic features.

## 1. Introduction

Cutaneous melanoma shows significant geographic variation in terms of incidence and mortality rates worldwide. According to the most recent global estimates, Australia has the highest age-standardized incidence rate, at around 37 cases per 100,000 people, yet it has a relatively low mortality rate of 2.3 deaths per 100,000 people. By contrast, the incidence rates in Europe and the United States are lower at an estimated 10.4 and 16.3 cases per 100,000 individuals, respectively, while the corresponding mortality rates are 1.5 and 1.1 deaths per 100,000 individuals [1,2].

The risk of developing melanoma arises from a complex interplay of genetic, phenotypic and environmental factors. Approximately 90% of melanoma cases are classified as sporadic, with the remaining 10% exhibiting hereditary components [3,4]. Most sporadic melanomas result from somatic variants induced by environmental factors, including ultraviolet (UV) radiation and aging [5,6]. These variants are confined to daughter cells and are not heritable [7]. An estimated 7–15% of melanoma cases occur in individuals with a family history, which may be influenced by shared environmental factors such as skin type, atypical moles and sun exposure [8,9]. Approximately 2% of melanomas are linked to pathogenic germline variants, with the risk of inheritance contingent upon genetic penetrance [10].

To enhance clarity regarding melanoma terminology, the following definitions are proposed, drawing on the insights of López Riquelme et al. [11]:
**Sporadic melanoma** (90% of all melanomas [3]): This term refers to a singular melanoma that arises in an individual without a family history of the disease. Most of these melanomas present somatic variants, while germline variants are identified in 1.2% to 11.1% of cases [12,13].**Multiple primary melanoma**: The definition of this term varies based on geographic region and local melanoma prevalence. In countries with low incidence rates, such as Italy and Spain, multiple primary melanomas refer to an individual developing two or more melanomas, either simultaneously or at different times (which constitutes 2% to 8% of all melanoma cases) [10,11,14]. In these cases, germline variants are present in 3.5% to 32% of instances [12,13].**Familial melanoma** (7–15% of all melanomas [15,16]): According to the International Melanoma Genetics Consortium, familial melanoma is defined by the presence of three or more melanoma cases within the same family branch or two or more cases among first-degree relatives [17]. Familial melanoma may be further classified into those with no detectable germline variants and those associated with low-risk or high-risk exhibiting high penetrance variants [10].**Hereditary melanoma** (2% of all melanomas; 20% of all familial melanomas): This category refers to sporadic or familial melanomas that are linked to high-risk genes with high penetrance [18].

## 2. Non-CDKN2A Germline Variants

Until recently, the primary genes that were recognized and clinically evaluated for predisposition to melanoma were CDKN2A/ARF (cyclin-dependent kinase inhibitor 2A) and its associated gene CDK4 (cyclin-dependent kinase 4) [15,19]. However, emerging research has highlighted the importance of additional non-CDKN2A genes, many of which are also occasionally mutated in melanoma cases [10] (Table 1).

Melanoma predisposition genes are classified based on their penetrance (designated as low, medium, or high), which reflects the probability that an individual carrying a specific gene variant will develop the disease. Although no identified germline variant provides a guarantee for the development of melanoma, the principal effect of these predisposition genes is to elevate the baseline risk, reducing the number of somatic variants necessary for oncogenesis [20].

**Table 1 diseases-13-00180-t001:** Non-CDKN2A variants, relative penetrance in low-melanoma-incidence countries, other tumor associations, and recommended screening.

Penetrance	Gene	Locus	Variants Most Frequently Associated	Other Tumor Associations Beyond Cutaneous Melanoma	Recommended Screening
High-Penetrance Genes	CDK4 (MIM# 123829)	12q14.1	*Missense* [21]: c.71G>A(p.R24H), c.70C>T(p.R24C)	Non-melanoma skin cancers; breast, pancreas, ovarian, cervical and stomach cancers; FAMMM syndrome [22,23,24,25].	Mole monitoring every 3, 6, or 12 months depending on risk factors; unclear recommendations for internal neoplasia screening [10,11].
	BAP1 (MIM# 606661)	3p.21.1	*Frameshift* [18]: c.1717del(L573Wfs*3)	Uveal melanoma; malignant mesothelioma; kidney, bladder, breast, and thyroid tumors; neuroendocrine tumors; basal cell carcinoma; rhabdoid meningiomas; paraganglioma; cholangiocarcinoma [26,27,28,29].	Mole monitoring every 6 months from age 18; ophthalmologic exam every 12 months from age 16; US or MRI chest, abdomen and urinary tract every 2 years; periodic physical examination of the abdomen and chest from age 30 [10,11].
	POT1 (MIM* 606478)	7q31.33	*Splicing* [19,30,31,32,33]: c.1687-1G>A(p.?) *Missense* [19,30,31,32,33]: c.266A>G(p.Y89C) c.280C>G(p.Q94E)c.255G>A(p.K85=)	Angiosarcoma; glioma; chronic lymphocytic leukemia; thyroid and colorectal cancers [30,33,34,35].	Mole monitoring every 12 months; annual full physical exam; complete blood count annually; US of the neck every 2 years; full-body MRI every 12 months ** [10,11].
	TERT (MIM* 187270)	5p15.33	*Promoter* [36]: c.-57T>G(p.?)	Chronic myeloproliferative neoplasms; bladder, colon, prostate, testicular, breast, kidney, and central nervous system tumors [37,38,39,40,41,42,43].	No sufficient evidence for clinical surveillance in these patients.
	ACD (MIM* 609377)	16q22.1	*Nonsense* [44]: c.958C>T(p.Q320*) *Missense* [44]: c.746A>G (p.N249S) *Frameshift* [44]: c.866_867delCT (p.P289Rfs*28)	Lymphoma B cells; leukemia; esophagus, breast, lung, kidney, cervical, colon, bladder, and prostatic cancers [22,45,46].	
	TERF2IP (MIM* 605061)	16q23.1	*Missense* [5,22,44]: c.572A>G(p.Q191R)		
Intermediate-Penetrance Genes	MC1R (MIM# 613099)	16q24.3	*Missense* [5]: c.252C>G(p.D84E) c.425G>A(p.R142H) c.451C>T(p.R151C), c.478C>T(p.R160W), c.880G>C(p.D294H)	Non-melanoma skin cancers [47].	No sufficient evidence for clinical surveillance in these patients.
	MITF (MIM# 614456)	3p13	*Missense* [10,48]: c.952G>A(p.E318K)	Pheochromocytoma; paragangliomas [49].	
	SLC45A2 (MIM# 227240)	5p13.2	*Missense* [50,51]: c.814G>A(p.E272K;rs26722), c.1368+697T>G(p.?; rs35407)		
Low-Penetrance Genes	IRF4 (MIM* 601900)	6p25.3		Lymphoma; leukemia and non-melanoma skin cancers [52].	
	ASIP (MIM# 611742)	20q13.33			
	TYR (MIM* 606933)	11q14.3			
	TYRP1 (MIM* 115501)	9p23			
	HERC2 (MIM* 605837)	15q13.1	*Non coding* [53,54]: c.*50G>A(p.?; rs1129038), c.13272+874T>G(p.?; rs12913832)	Uveal melanoma [55]; colorectal cancers [56].	
	NID1 (MIM* 131390)	1q21.3		Salivary adenoid cystic carcinoma and gliomas [57].	
	MTAP (MIM* 156540)	9p21		Colorectal cancer [58].	
	PLA2G6 (MIM* 603604)	22q13.1			
	ARNT (MIM* 126110)	1q21			
	SETDB1 (MIM* 604396)	1q21.3			
	PARP1 (MIM* 173870)	1q42.12	*Non coding* [59]: c.*526G>A(p.?; rs2695237)	Breast and ovarian cancers [60].	
	CYP1B1 (MIM* 601771)	2p22.2	*Missense* [59]: c.1294C>G(p.Leu432Val; rs1800440)	Breast and ovarian cancers [61].	
	CASP8 (MIM* 601763)	2q33.1	*Missense* [59]: c.652G>C(p.Asp218His; rs10931936)	Breast and esophagus cancers [60].	
	CDKAL1 (MIM* 611259)	6p22.3	*Missense* [62]: c.1226G>A(p.Arg409His; rs6914598)	Pancreatic cancer [63].	
	AGR3 (MIM* 615496)	7p21.1	*Non coding* [59]: c.*386G>A(p.?; rs117132860)	Breast and ovarian cancers [61].	
	TMEM38B (MIM* 611236)	9q31.2	*Non coding* [59]: c.*241C>T(p.?; rs10739220)	Breast cancer [61].	
	OBFC1 (MIM* 613129)	10q24.33	*Non coding* [59]: c.*30A>G(p.?; rs7902587)	Breast cancer [61].	
	CCND1 (MIM* 168461)	11q13.3	*Non coding* [59]: c.*7T>C(p.?; rs4354713)	Breast and head-neck cancers [64].	
	MX2 (MIM* 147559)	21q22.3	*Non coding* [59]: c.*63C>T(p.?; rs408825)	Lung cancer and leukemia [63].	
	MAFF (MIM* 604868)	22q13.1	*Non coding* [59]: c.*311G>A(p.?; rs132941)	Lung cancer [63].	
	FOXD3 (MIM* 602272)	1p31.3	*Non coding* [59]: c.*71C>T(p.?; rs670318)	Gliomas and neuroblastomas [65].	
	DSTYK (MIM* 611235)	1q32.1	*Non coding* [59]: c.*87G>A(p.?; rs2369633)	Breast and lung cancers [65].	
	DTNB (MIM* 607388)	2p23.3	*Non coding* [59]: c.*217G>A(p.?; rs12473635)	Breast cancer [65].	
	TERC (MIM* 602322)	3q26.2	*Non coding* [59]: c.*98C>T(p.?; rs3950296)	Lung cancer and leukemia [66].	
	GPR98 (MIM* 602851)	5q14.3	*Non coding* [59]: c.*269C>T(p.?; rs12523094)	Gliomas and breast cancer [66].	
	PPARGC1B (MIM* 608886)	5q32	*Non coding* [59]: c.*36G>A(p.?; rs32578)	Breast and ovarian cancers [66].	
	SOX6 (MIM* 607257)	11p15.2	*Non coding* [59]: c.*151A>G(p.?; rs7941496)	Liver cancer [67].	
	LMO3 (MIM* 606582)	12p13.2	*Non coding* [59]: c.*118C>T(p.?; rs4237963)	Neuroblastomas [68].	

** It should be considered in families that meet criteria similar to Li–Fraumeni syndrome starting at the age of 18.

### 2.1. High-Penetrance Non-CDKN2A Germline Variants

#### 2.1.1. CDK4

Germline variants in CDK4 genes are rare, with fewer than twenty families documented in the existing literature [22]. CDK4 (MIM# 123829) is an oncogene situated on chromosome 12 (12q14.1) that encodes a protein involved in the transition from the G1 phase to the S phase of the cell cycle [23,26]. Variants in CDK4 predominantly occur at codon 24, disrupting the interaction between CDK4 and p16INK4a. This disruption leads to aberrant cell cycle progression and tumor formation [26]. The most prevalent variants include c.71G>A (p.R24H) and c.70C>T (p.R24C) [21], while a less common variant associated with c.71G>T (p.R24L) has also been identified [23].

The clinical phenotype observed in families with CDK4 variants is similar to that of families with CDKN2A variants, exhibiting a significant presence of dysplastic nevi, along with an elevated risk of developing multiple primary melanomas. The average age at which individuals receive a diagnosis of melanoma is 39 years [22]. These melanomas frequently arise from precursor nevi, predominantly on the limbs, with superficial spreading melanoma being the most frequently encountered histological subtype [21,24].

Individuals who carry CDK4 variants may also face an increased risk of non-melanoma skin cancers, as well as cancers affecting the breast, pancreas, ovary, cervix and stomach [22]. In this contest, CDK4 variants are linked to FAMMM syndrome (Familial Atypical Multiple Mole Melanoma), which is characterized by multiple nevi, cutaneous and uveal malignant melanomas, and a high risk of pancreatic carcinoma [23,25]. Finally, in contrast to individuals with CDKN2A variants, those with CDK4 variants do not exhibit an increased risk for the development of central nervous system tumors, including astrocytomas [24].

#### 2.1.2. BAP1

The BAP1 gene (MIM# 606661) is implicated in approximately 1% of cutaneous melanomas and 4% of uveal melanomas [23]. Located on the short arm of chromosome 3 at region 3p21.1, this gene encodes a protein that exhibits carboxy-terminal ubiquitin hydrolase activity and serves as a binding partner for BRCA1 [22]. BAP1 is integral to several essential cellular processes, including cell division, signal transduction, and apoptosis, which occur through the mediation of the type 3 inositol 1,4,5-trisphosphate receptor (IP3R3) [69]. To date, researchers have identified 140 pathogenic variants of the BAP1 gene, which include 71 frameshift variants, 43 nonsense variants, 12 splice site variants, and 2 missense variants. Among these, the c.1717del(L573Wfs*3) variant is among the most prevalent, while a recently identified frameshift is noted as c.368delG (p.S 123Tfs*64) [18,70]. These variants are associated with uncontrolled cellular proliferation, significantly contributing to the predisposition to develop various types of tumors. The pronounced intrafamilial variability observed can be attributed to the loss of heterozygosity, which may result from a second variant, deletion, or epigenetic modifications [20,71].

It is estimated up to 90% of individuals carrying BAP1 variants will develop multiple red-orange papules or nodules, typically emerging in the second decades of life. These lesions referred to as BAP1-inactivated melanocytic tumors (BIMT) or BAPomas, are predominantly intradermal and demonstrate notable histopathological variability, ranging from epithelioid nevi to atypical melanocytic proliferations. Although classified as neoplasms of uncertain malignant potential, BAPomas are generally treated as melanoma cases [5,22,72] (Figure 1).

Five dermoscopic patterns have been described to occur in BIMTS: (1) structureless pink-to-tan with irregular dots/globules located eccentrically; (2) structureless pink-to-tan areas with peripheral vessels; (3) structureless pink-to-tan; (4) network with raised, structureless, pink-to-tan areas; and (5) globular pattern [10].

While the majority of BIMT cases are sporadic and exhibit an indolent course, their occurrence in the context of a germline BAP1 variant suggests a higher risk for a hereditary condition known as BAP1 tumor predisposition syndrome (BAP1-TPDS). This syndrome is associated with an elevated risk for various cancers, including cutaneous and uveal melanoma, malignant mesothelioma, renal cell carcinoma, basal cell carcinoma, and rhabdoid meningiomas. Additionally, other cancer associations include cholangiocarcinoma, breast cancer, thyroid cancer, bladder cancer, neuroendocrine tumors, and paragangliomas [26,27,28,29]. It is estimated that 63% of patients with germline BAP1 variants will develop at least one of these cancers before reaching the age of 53 [20]. Currently, there are no established diagnostic criteria for BAP1-TPDS. The syndrome should be suspected in individuals with two or more confirmed BAP1-TPDS tumors or one confirmed BAP1-TPDS tumor accompanied by a first- or second-degree relative with a confirmed BAP1-TPDS tumor (excluding basal cell carcinomas and cutaneous melanomas, which are prevalent in the general population). Diagnostic confirmation is achieved through the identification of a pathogenic heterozygous germline variant via molecular genetic testing [27].

#### 2.1.3. Genes of the Shelterin Complex

At the forefront of genomic protection is Shelterin, a telomere-specific protein complex that serves to safeguard chromosome ends. By facilitating a vital partnership between telomerase and telomeres, Shelterin plays an essential role in the maintenance of chromosome stability. Notably, germline variants in the genes encoding the proteins that comprise this complex can have significant implications for health and disease [73]. This analysis will explore the important functions of three key proteins: POT1, ACD, and TERF2IP.

POT1 (MIM* 606478), or Protection of Telomeres 1, is located on chromosome 7 and is recognized for its association with high-penetrance pathogenic variants [23]. These variants primarily affect the protein’s Oligonucleotide/Oligosaccharide-Binding (OB) domain [19,30,31]. A range of variants has been identified, including those that disrupt gene splicing (e.g., c.255G>A(p.K85=) and c.1687-1G>A) as well as missense variants (such as c.266A>G(p.Y89C) and c.280C>G(p.Q94E)), which ultimately alter the protein’s structure. These genetic alterations impede the ability of POT1 to effectively bind to single-stranded telomeric DNA, thereby hindering its elongation function [30,31,32]. Individuals harboring variants in POT1 are at an increased risk of developing multiple melanomas, particularly between the ages of 25 and 80 [33]. These skin cancers, frequently influenced by UV exposure, are characterized by distinctive spitzoid morphology and an increased lymphocytic infiltrate [74]. In addition, variants within this gene are associated with a spectrum of other malignancies, including angiosarcoma, glioma, chronic lymphocytic leukemia, thyroid carcinoma, and colorectal cancer [30,33,34,35] (Figure 2).

The next protein, ACD, known as adrenocortical dysplasia protein homolog, contains a domain that binds to POT1, significantly enhancing its affinity for telomeric DNA [75].

ACD (MIM* 609377) is also crucial in linking the Shelterin complex with TERT, a pivotal telomerase component that regulates its activity at telomeres. Noteworthy variants in the ACD gene, including c.958C>T(p.Q320*) (a nonsense variant), c.746A>G(p.N249S) (a missense variant), and c.866_867delCT (p.P289Rfs*28) (a frameshift variant), have considerable implications for cellular health [44].

Lastly, TERF2IP (MIM* 605061), or telomeric repeat-binding factor 2-interacting protein, is a critical player that interacts with various factors to silence subtelomeric genes, thus contributing to telomere stability. Variants in the TERF2IP gene, which include a nonsense variant and several missense variants such as c.572A>G(p.Q191R), have been linked to vulnerabilities in cellular integrity [5,22,44].

Similarly to variants in POT1, alterations in ACD and TERF2IP increase the risk of multiple primary melanomas, which are typically diagnosed at a young age, with incidence peaking in the second decade of life. These melanomas predominantly manifest as either superficial spreading or lentigo maligna types, though spitzoid variants have also been documented. Furthermore, variants in these genes exacerbate the risk for a variety of other malignancies, including hematological cancers such as B-cell lymphoma and leukemia, which may lead to severe complications such as bone marrow failure and aplastic anemia. The spectrum of associated malignancies also encompasses cancers of the esophagus, breast, lung, kidney, cervix, ovary, colon, urinary tract, and prostate [22,45,46].

#### 2.1.4. TERT (Telomerase Reverse Transcriptase)

The TERT gene (MIM* 187270), located in the 5p15.1 region of chromosome 5, encodes the catalytic subunit of telomerase, an enzyme essential for maintaining telomere length and chromosomal stability [76]. Variants in the TERT promoter are associated with excessive telomere elongation, which can uncontrollably activate telomerase, promoting cell survival and the accumulation of oncogenic variants, thus contributing to cancer development. Somatic variants in TERT are common in melanoma, with an incidence of 68% in primary melanomas [44,76]. Germline variants in TERT are rare, with c.-57T>G being the most frequent [36].

This variant is associated with early-onset melanomas, occurring between the ages of 18 and 46, with an aggressive clinical course and unfavorable prognosis, characterized by a high mortality rate with death occurring within three years of diagnosis [22,36,77]. Carriers of variants in the TERT gene tend to develop a large number of nevi with a predominantly globular pattern, typically visible on dermoscopy [78] (Figure 3).

Variants in the TERT gene are responsible for approximately 1% of familial melanoma cases and increase the risk of familial liver diseases, idiopathic pulmonary fibrosis, and lung cancer. Furthermore, these variants are associated with an increased risk of chronic myeloproliferative neoplasms and cancers at various sites, including the bladder, colon, prostate, testes, breast, kidneys, and central nervous system [37,38,39,40,41,42,43].

### 2.2. Intermediate-Penetrance Non-CDKN2A Germline Variants

#### 2.2.1. MC1R

The MC1R gene (MIM# 613099) is located on chromosome 16q24 and exhibits an autosomal recessive inheritance pattern [10,79]. It encodes the melanocortin 1 receptor (MC1R), which is categorized as a G protein-coupled receptor. Upon activation by α-melanocyte-stimulating hormone (α-MSH) or UV radiation, this receptor initiates a signaling cascade that promotes the proliferation of melanocytes and the biosynthesis of melanin. This pathway is crucial for the production of eumelanin (brown/black pigment) and plays a significant role in regulating skin and hair pigmentation [22,23]. The MC1R gene is characterized by a high degree of polymorphism, with 79 identified variants [80], many of which are relatively prevalent, affecting approximately 11% of the population [22]. The predominant alleles, known as R alleles (for example, c.252C>G(p.D84E)), are distinguished by diminished expression of the receptor at the cell surface. This reduced receptor expression contributes to a decrease in eumelanin synthesis, leading to an increase in the production of pheomelanin (red/yellow pigment) and resulting in the Red Hair Color (RHC) phenotype. This phenotype is associated with red hair, fair skin, freckles, and a reduced tanning response. Although most studies have not directly established a correlation between the MC1R and eye color, one specific R variant (c.425G>A(p.R142H)) has been linked to the presence of blue and green eyes [81]. Furthermore, individuals possessing R alleles of the MC1R exhibit a 28% increased risk of developing melanoma, independent of other phenotypic factors [82]. Although MC1R variants increase susceptibility to melanoma, they do not fully explain individual risk. The M-SKIP project showed that there were no significant additive or multiplicative interactions between MC1R variants and general measures of sun exposure. However, a significant additive interaction was identified specifically between MC1R R alleles and a high number of sunburns. This suggests that acute sun damage may play a critical modulatory role in melanoma risk among individuals carrying these variants [83]. Additionally, R alleles in MC1R have been implicated in an increased cancer risk for individuals harboring variants in the CDKN2A gene [84].

R alleles of MC1R are associated with specific dermatological characteristics, including blue nevi, hypopigmented nevi larger than 2 mm in diameter (commonly termed white nevi), and larger congenital melanocytic nevi, without an overall increase in the number of nevi [85]. Dermoscopic examination of these nevi reveals structural instability and a lighter hue, frequently exhibiting a distinctive vascular pattern [86,87]. In the context of melanoma, carriers of MC1R R alleles tend to present with larger, hypopigmented melanomas, often described as red melanomas. These lesions exhibit dermoscopic features characterized by reduced pigmentation, compromised structural integrity, atypical features, and heightened vascularity. Consequently, these melanomas typically present with lower ABCD Dermoscopy (TDS) scores compared to individuals without the RHC phenotype [88]. The extent of diminished pigmentation and structural complexity correlates with the number of R variants present. Histological analyses of these melanomas often reveal increased tumor thickness, mitotic activity, ulceration, and the presence of tumor-associated lymphocytes [89]. These melanomas are predominantly localized to the upper body, particularly the upper limbs [88] (Figure 4).

Additionally, carriers of MC1R variants who do not display the RHC phenotype may nonetheless experience an elevated risk of developing non-melanoma skin cancers, including squamous cell carcinoma (SCC) and basal cell carcinoma (BCC). This risk is further amplified in individuals possessing a higher number of pathogenic MC1R variants [47].

#### 2.2.2. MITF

The MITF gene (MIM# 614456) (microphthalmia-associated transcription factor), located on chromosome 3, belongs to the Myc gene family. It encodes a protein that acts as a key regulator of pigmentation, differentiation, and survival of melanocytes. Its function is modulated by SUMO proteins (small-ubiquitin-like modifier proteins) [90]. The most common germline variant of the MITF, c.952G>A(p.E318K), is present in 1% of individuals of European descent. This missense variant, which replaces glutamic acid with lysine at position 318, alters the MITF’s affinity for the SUMO protein, reducing its SUMOylation. As a result, it increases the MITF’s transcriptional activity, leading to dysregulated cell proliferation, suggesting a potential oncogenic role [10,48]. The germline E318K variant is associated with an increased risk of developing melanoma, which may be 3 to 5 times higher than in the general population. This risk further increases, from 8 to 31 times, in individuals with a positive family history of pancreatic cancer or renal carcinoma, respectively [91].

Additionally, the p.E318K variant of the MITF gene is also correlated with a higher risk of developing pheochromocytomas and paragangliomas [49].

MITF-mutated patients present distinctive phenotypic characteristics, including dark hair, fair skin, non-blue eyes, and a high number of pink or light brown nevi, generally with a diameter greater than 5 mm [48,92]. These nevi show an atypical reticular dermoscopic pattern, suggesting photoinduced nevogenesis [93]. Furthermore, these patients are more likely to develop multiple primary melanomas, with onset before the age of 40 [92]. The typical locations of melanomas include the back, followed by the legs, arms, and abdomen (areas intermittently exposed to UV radiation) [20]. These melanomas are often of the nodular melanotic or hypomelanotic type and exhibit atypical polymorphic vessels [78] (Figure 5). Histopathologically, they tend to be thicker, associated with a worse prognosis, as their growth rate is higher than 0.4 mm per month, compared to the average 0.1 mm per month observed in common melanomas [22,92].

#### 2.2.3. SLC45A2

SLC45A2 (MIM# 227240), a gene located on chromosome 5, is described in many studies as a gene with intermediate penetrance [5,23], while others consider it to have low penetrance [18]. It encodes a 530-amino acid membrane protein that interacts, through a mechanism still partially unknown, with a transporter associated with the melanosome, the cellular organelle responsible for melanin production. Some single-nucleotide polymorphisms of SLC45A2, such as rs26722 (c.814G>A(p.E272K)) and rs35407 (c.1368+697T>G), have been suggested as risk factors for the development of melanoma [50,51] (Figure 6).

### 2.3. Low-Penetrance Non-CDKN2A Germline Variant

Low-penetrance genes potentially associated with melanoma are involved in various biological functions (Table 1).

The first group of genes is involved in the regulation of skin and hair pigmentation. Variants of these genes, which result in lighter pigmentation, increase susceptibility to melanoma by reducing protection against UV radiation damage. These include the following:ASIP (Agouti Signaling Protein) (MIM# 611742): encodes an antagonist of α-MSH that binds to MC1R, inhibiting eumelanin synthesis [94];TYR (MIM* 606933): encodes the enzyme tyrosinase, which catalyzes the first two steps in melanin synthesis [51,95];TYRP1(MIM* 115501): encodes a protein that acts as a cofactor for tyrosinase, influencing the tanning response [95];OCA2 (MIM* 605837): encodes a membrane protein involved in melanin synthesis. Some single-nucleotide polymorphisms (rs1129038(c.*50G>A), rs12913832(c.13272+874T>G)) in the HERC2/OCA2 region are associated with eye color and increase the risk of melanoma [53,54,96] (Figure 7).

A second group of genes, whose precise role remains unclear, is involved in the development of nevi: NID1 (MIM* 131390), MTAP (MIM* 156540), and PLA2G6 (MIM* 603604) [22,57,97,98,99,100].

A third group of genes, including IRF4 (MIM* 601900), is associated with both pigmentation and nevus formation [101,102].

Genes not directly related to pigmentation and nevus density but with important roles in melanoma have also been identified, such as ARNT (MIM* 126110) (involved in the response to xenobiotics) and SETDB1 (MIM* 604396) (involved in immune cell development) [103,104,105,106].

Finally, some low-penetrance genetic variants associated with multiple primary melanomas have been identified in genes like CLPTM1L, NCOA6, MX2, and a haplogroup of PARP1 [107].

## 3. Other Genes Associated with Hereditary Cancer Syndromes

Some gene variants are associated with the development of multiple tumors. Although melanoma is not the most frequently identified neoplasm linked to these genetic syndromes, individuals with such variants exhibit a heightened risk of developing melanoma in comparison to the general population (Table 2) [18].

## 4. Genetic Test

Current clinical guidelines do not recommend routine genetic testing for all melanoma patients; instead, testing is advised when there is a greater than 10% likelihood of a predisposing variant. This decision is typically based on factors such as the patient’s family history, geographic factors, age at melanoma onset, and associations with other types of neoplasms [123]. The 2023 Italian guidelines recommend testing for the CDKN2A gene in individuals with multiple melanoma diagnoses or a significant family history of the condition. Since susceptibility to melanoma is not solely determined by CDKN2A, genetic testing is also suggested in the following scenarios:Patients with melanoma who have a positive family history (at least two affected family members) or a personal history suggestive of multiple melanoma occurrences;Patients with melanoma who have a family or personal history of other cancers, such as pancreatic adenocarcinoma, uveal melanoma, mesothelioma (pleural or peritoneal), or renal cancers;Patients with a history of atypical Spitz nevus excisions or those who meet the diagnostic criteria for syndromes associated with increased melanoma risk, such as hamartomatous syndromes with PTEN variants or BAP1 syndrome.

Genetic testing typically involves a multigene panel that includes genes such as CDKN2A, CDK4, BAP1, POT1, TERF2IP, ACD, TERT, MITF, MC1R, ATM, and PALB2. In addition, variants in other key genes like TP53, RB1, PTEN, and BRCA2 may also be evaluated [124].

The most commonly employed technique for genetic sequencing is polymerase chain reaction (PCR), which amplifies specific DNA sequences to detect a limited range of variants. Recently, next-generation sequencing (NGS) has revolutionized molecular diagnostics, enabling the sequencing of millions of DNA fragments simultaneously and providing the ability to analyze entire genomes or specific regions in great detail [125]. While PCR is effective for detecting particular variants, NGS offers a broader and more thorough analysis, detecting a wider spectrum of genetic variations. However, PCR remains faster, more affordable, and simpler compared to NGS. Some specialists recommend using PCR as an initial screening tool to exclude patients with known variants, reserving NGS for identifying additional, relevant variants [11].

It is important to note that not all genetic variants identified are pathogenic; many show variable expression and incomplete penetrance. Variants are classified using the standardized terminology set by the American College of Medical Genetics and Genomics, which includes the following:Pathogenic—Class 1;Likely pathogenic—Class 2;Variants of uncertain significance—Class 3;Likely benign—Class 4;Benign—Class 5 [126].

While the identification of germline variants in patients does not immediately alter clinical management, it serves to guide the families of affected individuals, who are at higher risk for the disease, toward appropriate primary and secondary prevention strategies [124] (Table 1).

In parallel with these targeted genetic approaches, polygenic risk scores (PRSs) have emerged as a complementary and promising method for estimating genetic susceptibility to cutaneous melanoma. The PRS combines the cumulative effect of several single-nucleotide polymorphisms (SNPs), thereby improving risk stratification, even for individuals lacking high-penetrance pathogenic variants in genes such as CDKN2A and BAP1. Integrating SNPs into predictive models alongside traditional clinical and environmental factors has been shown to significantly improve the identification of high-risk individuals. Key clinical applications include more precise genetic risk stratification in the absence of overt phenotypic features, personalization of prevention and surveillance strategies through targeted screening programs, and supporting prognosis and therapeutic management thanks to PRS’s ability to provide additional information on disease progression and treatment response. However, further validation studies are required in geographically and ethnically diverse cohorts to ensure the robustness of PRSs and their effective application in medical practice [127,128,129].

## 5. Discussion

In recent years, significant progress has been made in melanoma research with the identification of novel germline variants involved in melanoma susceptibility. While the CDKN2A gene remains the primary high-penetrance locus associated with familial melanoma, several recent studies have revealed the presence of additional non-CDKN2A germline variants that may be implicated in individual susceptibility. This review focuses on these non-CDKN2A variants, summarizing the most significant articles available in widely used scientific databases such as PubMed, Web of Science, and the Cochrane Library.

The identification of these variants has been made possible by advancements in sequencing technologies, particularly next-generation sequencing (NGS), which expand diagnostic possibilities beyond single-gene approaches [11,12,13,14,15,16,17,18,19,20,21,22,23,24,25,26,27,28,29,30,31,32,33,34,35,36,37,38,39,40,41,42,43,44,45,46,47,48,49,50,51,53,54,57,69,71,72,73,74,75,76,77,78,79,80,81,82,83,84,85,86,87,88,89,90,91,92,93,94,95,96,97,98,99,100,101,102,103,104,105,106,107,108,109,110,111,112,113,114,115,116,117,118,119,120,121,122,123,124,125,130]. Some of these variants have been associated with complex neoplastic syndromes that include other tumor forms in addition to melanoma, confirming the pleiotropic nature of many of the genes involved and the complexity of their contribution to pathogenesis.

One issue that arose from this review is the variability in penetrance associated with each variant. Genetic penetrance is defined as the probability that an individual carrying a given mutation will exhibit a phenomenon modulated by multiple factors, including genetic, epigenetic, environmental, and stochastic factors. This means that mutations in other susceptibility genes, as well as individual factors such as age, sex, and lifestyle, contribute to modulating the probability of developing the disease. This highlights that penetrance is a dynamic phenomenon influenced by multiple elements besides the genetic variant alone. This allows us to conclude that a crucial yet often overlooked aspect of melanoma risk assessment is the complex interaction between genetic variants and environmental factors, particularly ultraviolet (UV) exposure. Our review showed that variants such as MC1R increase genetic predisposition and reduce the skin’s ability to defend itself against UV damage. This may explain why individuals with clear phototypes develop melanomas after repeated sunburn, even in the absence of high-penetrance mutations [83].

For the same reason, alongside the analysis of high-penetrance genes, polygenic risk scores (PRSs) are becoming more widespread. PRSs are tools that sum up the effect of many low-penetrance genetic variants (SNPs) and offer a more complete risk profile. PRSs are particularly useful for identifying individuals at risk, even when there is no obvious mutation in major genes [127,128,129].

The interaction between skin genotype and phenotype can result in unusual clinical and dermoscopic presentations of nevi. This makes it possible to identify patterns that are specific to certain genetic arrangements. Significant advances in dermoscopy, which now have increasingly sophisticated instruments, have also made this possible. A major challenge lies in integrating genetic information with visible phenotypic traits, particularly the dermoscopic features of nevi, to support dermatologists in identifying patients at risk early on. For instance, variants of the MC1R gene, which are prevalent among light phototypes, are linked to poorly pigmented melanomas that are challenging to identify due to their poor pigmentary structures and atypical vascular patterns when examined using dermoscopy [81]. Similarly, carriers of the MITF p mutation exhibit these characteristics. Those with the E318K mutation have dark hair, fair skin, non-blue eyes, and a high number of pink or light brown nevi with an atypical reticular dermoscopic pattern, as well as melanotic or hypomelanotic nodular melanomas with atypical polymorphic vessels [10,48,92,93].

Integrating genetic analysis with phenotypic and dermoscopic observations thus enables melanomas to be recognized at an early stage, even when they present as ‘mini-melanomas’, i.e., lesions that are small in size but already potentially malignant [131].

Meanwhile, artificial intelligence (AI) is playing an increasingly important role in the early detection of melanoma. AI algorithms applied to dermoscopy are proving effective in recognizing suspicious lesions, even in patients who carry genetic variants and have numerous dysplastic nevi. In the future, such tools could provide valuable support for the early screening of genetically predisposed individuals, improving diagnostic accuracy and sensitivity while reducing interobserver variability [62].

One limitation of the existing literature is the underrepresentation of Mediterranean populations in genetic melanoma studies. As Pellegrini et al. [132] pointed out, most research has focused on populations from northern Europe, the US, and Australia, neglecting the genetic characteristics peculiar to southern Europe. In an attempt to address this disparity, the MelaNostrum Consortium conducted an extensive study analyzing genetic variants associated with melanoma susceptibility in families from the Mediterranean region. This study took an innovative approach, combining the examination of clinical features with the analysis of genetic variants in a large group of Mediterranean families. This synergy enabled the identification of distinct hereditary melanoma traits that are characteristic of this culturally diverse region. This study represents a significant advance in melanoma research, providing valuable information for the future classification of risk based on ethnic and geographical background [133].

## 6. Conclusions

The future of melanoma management is moving towards a more holistic approach that combines genetic analysis with phenotypic and dermoscopic observations and an assessment of environmental factors. Continued advances in genetic testing now enable the identification of germline variants with variable penetrance that are increasingly tailored to the specific characteristics of different global populations. Meanwhile, dermoscopic examination enables the identification of morphological patterns indicative of specific genetic predispositions, thereby improving the early detection of individuals at risk.

A thorough understanding of the interplay between genetic variants and environmental factors is therefore crucial for refining risk stratification models. This will enable the development of more effective screening protocols, even for less characterized variants, and, most importantly, the implementation of truly personalized prevention and surveillance strategies. This holistic approach is a significant step towards the early and accurate diagnosis of melanoma and related malignancies, offering substantial benefits in terms of prevention and clinical management.

## Figures and Tables

**Figure 1 diseases-13-00180-f001:**
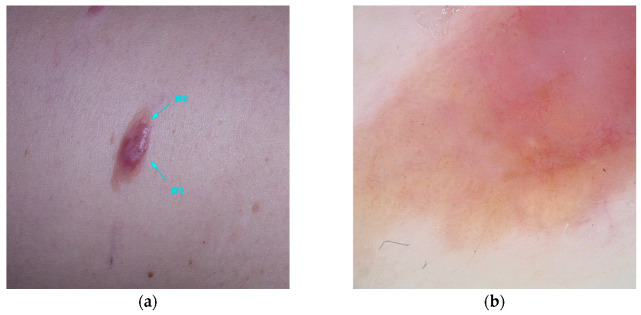
Clinical (**a**) and dermoscopic (**b**) image of a BAP1 melanoma located on the abdomen.

**Figure 2 diseases-13-00180-f002:**
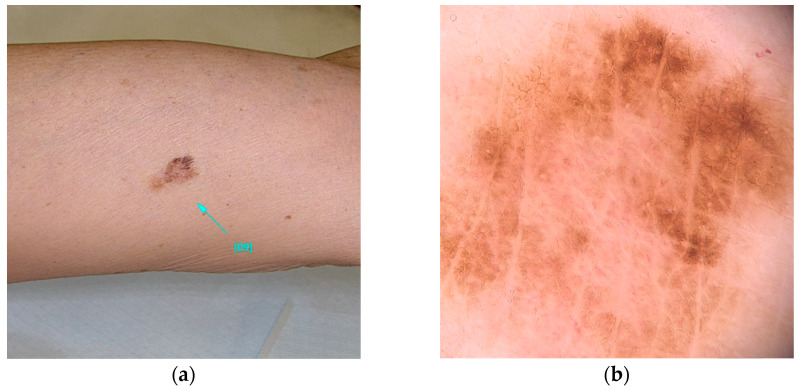
Clinical (**a**) and dermoscopic (**b**) image of a POT1 melanoma located on the left arm.

**Figure 3 diseases-13-00180-f003:**
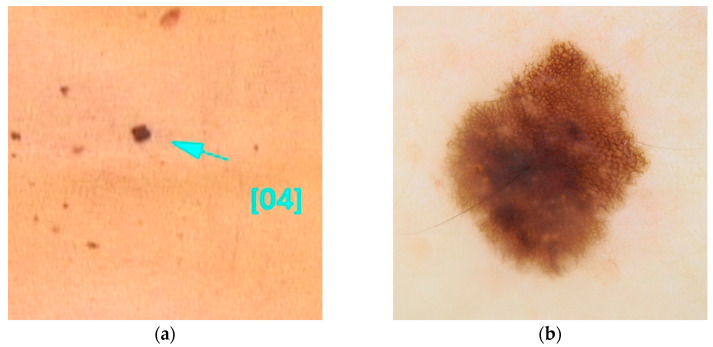
Clinical (**a**) and dermoscopic (**b**) image of a TERT melanoma located on the back.

**Figure 4 diseases-13-00180-f004:**
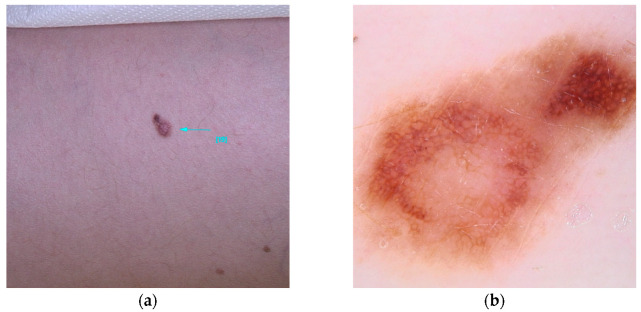
Clinical (**a**) and dermoscopic (**b**) image of an MC1R melanoma located on the left arm.

**Figure 5 diseases-13-00180-f005:**
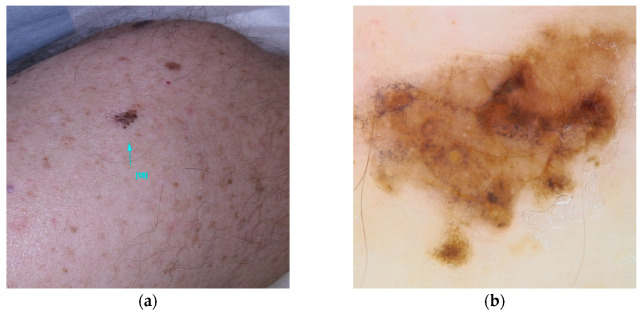
Clinical (**a**) and dermoscopic (**b**) image of a MITF melanoma located on the left arm.

**Figure 6 diseases-13-00180-f006:**
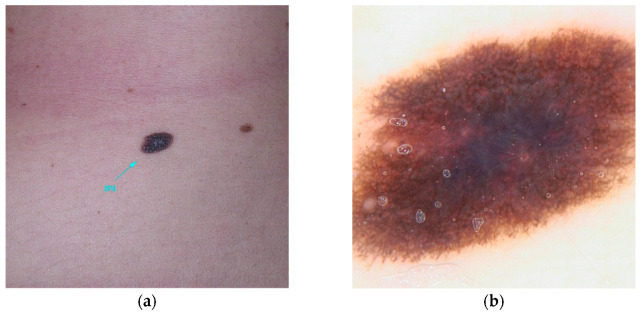
Clinical (**a**) and dermoscopic (**b**) image of a SLC45A2 melanoma located on the abdomen.

**Figure 7 diseases-13-00180-f007:**
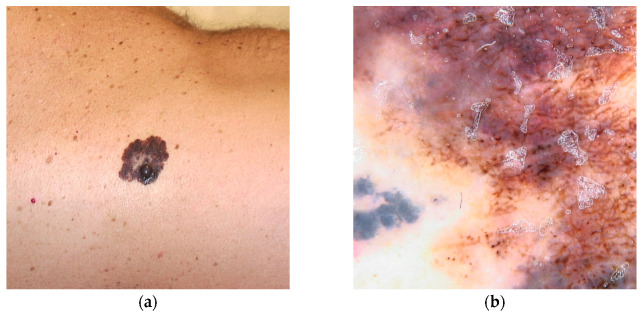
Clinical (**a**) and dermoscopic (**b**) image of an OCA2 melanoma located on the right flank.

**Table 2 diseases-13-00180-t002:** Some hereditary syndromes with an increased risk of melanoma.

Gene	Hereditary Syndrome	Locus	Other Associated Cancers
BRCA1 (MIM* 113705)	Hereditary Breast and Ovarian Cancer Syndrome (HBOC)	17q21.31	Breast, ovarian, prostate, and pancreatic cancers [108].
BRCA2 (MIM* 600185)		13q13.1	
TP53 (MIM# 151623)	Li–Fraumeni Syndrome	17p13.1	Breast, colorectal, pancreas, central nervous system, ovary, adrenal, and lung cancers; soft tissue sarcomas and bone tumors; hepatocellular carcinoma and lymphoma [109].
CHEK2 (MIM# 609265)	CHEK2-Associated Cancer Predisposition Syndrome	22q12.1	Osteosarcoma; breast, colorectal, and prostate cancers [110].
PTEN (MIM# 158350)	Cowden Syndrome	10q23.31	Breast, thyroid, kidney, colorectal, and endometrial cancers; glioma [111].
RB1 (MIM# 180200)	Retinoblastoma Syndrome	13q14.2	Retinoblastoma; osteosarcoma; bladder, lung, and pineal gland tumors [112].
FLCN (MIM# 135150)	Birt–Hogg–Dubé Syndrome	17p11.2	Kidney and lung cancers [24].
PTCH1 (MIM* 601309)	Gorlin Syndrome	9q22.32	Basal cell carcinoma; sarcomas; medulloblastoma; meningioma; pancreas, breast, lung, colon, and ovarian cancers [113].
SUFU (MIM* 607035)		10q24.32	
WRN (MIM# 277700)	Werner Syndrome	8p12.3	Osteosarcoma and other sarcomas; thyroid cancer; meningioma and leukemia [114].
MLH1 (MIM* 120436)	Lynch Syndrome	3p22.2	Colorectal, endometrial and urothelial cancers; gastrointestinal tumors [115].
MSH2 (MIM * 609309)		2p21-p16	
MSH6 (MIM* 600678)		2p16.3	
PMS2 (MIM# 614337)		7p22.1	
STK11 (MIM# 175200)	Peutz–Jeghers Syndrome	19p13.3	Colon, stomach, esophagus, small intestine, pancreas, ovary, testicle, and thyroid cancers [116].
ATM (MIM# 208900)	Ataxia–telangiectasia	11q22.3	Breast, pancreas, prostate, stomach, ovarian, colorectal, and cancers; hematologic malignancies [117].
NF1 (MIM# 162200)	Neurofibromatosis Type 1	17q11.2	Myelomonocytic leukemia; lymphomas; endocrine tumors; sarcoma; pheochromocytoma; breast, thyroid, bone, ovarian, and central nervous system tumors; gastrointestinal tumors [118].
ERCC2 (MIM* 126340)	Xeroderma Pigmentosum	19q13.32	Non-melanoma skin cancer; lung, brain, breast, stomach, pancreas, colorectal, and urogenital tract cancers; leukemia and sarcomas [119,120,121,122].
ERCC3 (MIM* 133510)		2q14.3	
ERCC4 (MIM* 133520)		16p13.12	
XPA (MIM* 611153)		9q22.33	
XPC (MIM* 613208)		3p25.1	

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
