# Peer review of "Germline Non-CDKN2A Variants in Melanoma and Associated Hereditary Cancer Syndromes"

_diseases, 2025, doi:10.3390/diseases13060180_

Round 1
Reviewer 1 Report
Comments and Suggestions for Authors
This is a rich and promising manuscript with clear clinical relevance and a well-chosen focus on non-CDKN2A variants. Integration of clinical, dermoscopic, and genetic perspectives is valuable and up-to-date. No adjustments required
Author Response
We thank the reviewer for the positive feedback on our work
Reviewer 2 Report
Comments and Suggestions for Authors
This is a review on inherited non-CDKN2A variants in melanoma and associated cancer syndromes, providing also information on genotype-phenotype correlation. The article is well-written, concise, and informative.
Some minor comments:
- Introduction, “In high incidence countries, annual mortality is estimated at 1,500 to 2,000 deaths, whereas Europe and the United States experience mortality rates of approximately 1 to 3 deaths per 100,000 individuals [2].”: It would be preferable to use data from an international database, such as Globocan.
- Table 1 and 3. Low Penetrance non-CDKN2A Germline Variant: 1) Apart from the low-penetrance genes listed, there are several others. The authors could take into consideration the article by Landi et al (Nat Genet. 2020; 52(5): 494–504. doi:10.1038/s41588-020-0611-8).
- 2.1. MC1R, “Nevertheless, the presence of MC1R variants does not fully explain the risk of melanoma, underscoring the critical influence of environmental factors, particularly UV exposure, in modulating this risk among individuals with RHC variants [62]”: The MC1R - sun exposure interaction is investigated in an article at J Eur Acad Dermatol Venereol. 2024. doi: 10.1111/jdv.20380. There were no additive or multiplicative interactions between MC1R variants and any sun-related measure, while a significant additive interaction was observed for MC1R R variants and high number of sunburns.
- Genetic Test: The authors should also comment on the potential role of polygenic risk scores (PRS) in further defining genetic susceptibility for melanoma.
Author Response
- We thank the reviewer for the suggestion, we revised the body of the text by making the suggested changes and updating the age-standardised ASR (worldwide) rates based on the latest estimates provided by GLOBOCAN. In particular, we have made the text more readable and included data for Australia, Europe and the United States:
“ Cutaneous melanoma shows significant geographic variation in terms of incidence and mortality rates worldwide. According to the most recent global estimates, Australia has the highest age-standardised incidence rate, at around 37 cases per 100,000 people, yet it has a relatively low mortality rate of 2.3 deaths per 100,000 people. By contrast, the incidence rates in Europe and the United States are lower at an estimated 10.4 and 16.3 cases per 100,000 individuals respectively, while the corresponding mortality rates are 1.5 and 1.1 deaths per 100,000 individuals [1,2].” - We thank the reviewer for the suggestion. Having read the recommended article, we have added the missing low-penetrance genes to Table 1. We have associated an MIM number with each of them, except for the genes marked with ***, for which there is no MIM number, as specified in the appendix to the table. We have also associated the gene locus, identified variants, and other tumours associated with the aforementioned sources. Additionally, we have added some tumour associations for genes already included in the table, as well as a reference to Table 1 in the sub-section 'Low Penetrance Non-CDKN2A Germline Variant'. All of the aforementioned changes are marked in red in the body of the text.
-
We thank the reviewer for the comment and clarification. We have amended the text to specify that the M-SKIP project has demonstrated that acute sun damage may play a critical modulatory role in melanoma risk among individuals carrying the MC1R R allele. We have also added the recommended article to the bibliography (62).
- We thank the reviewer for the suggestion. Of the various publications on polygenic risk scores, we selected the following three:
- Cust, A. E., Drummond, M., Kanetsky, P. A., Mann, G. J., Schmid, H., Hopper, J. L., Aitken, J. F., Armstrong, B. K., Giles, G. G. and Holland, E. (2018). 'Evaluation of the incremental contribution of common genomic variants to melanoma risk prediction in two population-based studies'. Journal of Investigative Dermatology, 138(12), pp.2617–2624.
- Wong, C. K., Dite, G. S., Spaeth, E. A., Murphy, N. M., & Allman, R. (2018). 'Melanoma risk prediction based on a polygenic risk score and clinical risk factors'. Melanoma Research, 33(4): p. 293–299, August 2023.
-Julia Steinberg, Mark M. Iles, Jin Yee Lee, Xiaochuan Wang, Matthew H. Law, Amelia K. Smit, Tu Nguyen-Dumont, Graham G. Giles, Melissa C. Southey, Roger L. Milne et al., 'Independent evaluation of melanoma polygenic risk scores in UK and Australian prospective cohorts', British Journal of Dermatology, Vol. 186, No. 5, 1 May 2022, pp. 823–834.After this paragraph, we have supplemented the argument as follows: “In parallel with these targeted genetic approaches, polygenic risk scores (PRS) have emerged as a complementary and promising method for estimating genetic susceptibility to cutaneous melanoma. PRS combines the cumulative effect of several single-nucleotide polymorphisms (SNPs), thereby improving risk stratification, even for individuals lacking high-penetrance pathogenic variants in genes such as CDKN2A and BAP1. Integrating SNPs into predictive models alongside traditional clinical and environmental factors has been shown to significantly improve the identification of high-risk individuals. Key clinical applications include more precise genetic risk stratification in the absence of overt phenotypic features, personalisation of prevention and surveillance strategies through targeted screening programmes, and supporting prognosis and therapeutic management thanks to PRS's ability to provide additional information on disease progression and treatment response. However, further validation studies are required in geographically and ethnically diverse cohorts to ensure the robustness of PRSs and their effective application in medical practice”.
Reviewer 3 Report
Comments and Suggestions for Authors
This is a well written review article evaluating non-CDKN2A germline variants and their dermoscopic and phenotypic features.
Discussion section is not clear. I would recommend that authors rewrite this section in and extend it explaining the link between genetic aspects and dermoscopic/phenotypic features. Gene-environment interaction should be properly addressed in this section.
Also, I suggest a short Conclusions section.
Author Response
We thank the reviewer for the suggestion. Following suggestions received, we have rewritten the discussion section, aiming to make it clearer and more coherent. Specifically, we have highlighted in the opening section how recent advances in genetic sequencing techniques have made it possible to identify numerous non-CDKN2A germline variants associated with melanoma susceptibility, including those linked to complex neoplastic syndromes. We then delved into the concept of penetrance, emphasising its variability and the various factors that influence it, such as genetic, environmental and individual characteristics. We also incorporated a brief description of Polygenic Risk Scores, as recommended by Reviewer 1. We then focused on the interactions between genetics and the environment, as well as the correlation between genotype, skin phenotype and dermoscopic features. We emphasised how the latter can facilitate early diagnosis. We also addressed the significant underrepresentation of Mediterranean populations in the current literature and described the MelaNostrum Consortium's contribution to addressing this disparity. Finally, as requested, we have added a concluding section emphasising the importance of taking a holistic approach that integrates genetics, phenotype, dermoscopy and environmental factors in order to improve primary and secondary prevention strategies for melanoma.
Round 2
Reviewer 3 Report
Comments and Suggestions for Authors
Satisfied with authors answer.